# Tracking of Infused Mesenchymal Stem Cells in Injured Pulmonary Tissue in *Atm*-Deficient Mice

**DOI:** 10.3390/cells9061444

**Published:** 2020-06-10

**Authors:** Patrick C. Baer, Julia Sann, Ruth Pia Duecker, Evelyn Ullrich, Helmut Geiger, Peter Bader, Stefan Zielen, Ralf Schubert

**Affiliations:** 1Division of Nephrology, Department of Internal Medicine III, University Hospital, Goethe-University, 60596 Frankfurt am Main, Germany; julia.sann@t-online.de (J.S.); h.geiger@em.uni-frankfurt.de (H.G.); 2Division for Allergy, Pneumology and Cystic Fibrosis, Department for Children and Adolescents, University Hospital, Goethe-University, 60596 Frankfurt am Main, Germany; ruthpia.duecker@kgu.de (R.P.D.); stefan.zielen@kgu.de (S.Z.); 3Division of Pediatric Stem Cell Transplantation and Immunology, Department for Children and Adolescents Medicine, University Hospital Frankfurt, Goethe University, 60596 Frankfurt am Main, Germany; evelyn.ullrich@kgu.de (E.U.); peter.bader@kgu.de (P.B.); 4Experimental Immunology, Department for Children and Adolescents Medicine, University Hospital Frankfurt, Goethe University, 60596 Frankfurt am Main, Germany; 5German Cancer Consortium (DKTK) partner site Frankfurt/Mainz, 60596 Frankfurt am Main, Germany

**Keywords:** tracking, mesenchymal stromal/stem cells, bio imaging, bioluminescence, qRT-PCR, Ataxia telangiectasia, Atm

## Abstract

Pulmonary failure is the main cause of morbidity and mortality in the human chromosomal instability syndrome Ataxia-telangiectasia (A-T). Major phenotypes include recurrent respiratory tract infections and bronchiectasis, aspiration, respiratory muscle abnormalities, interstitial lung disease, and pulmonary fibrosis. At present, no effective pulmonary therapy for A-T exists. Cell therapy using adipose-derived mesenchymal stromal/stem cells (ASCs) might be a promising approach for tissue regeneration. The aim of the present project was to investigate whether ASCs migrate into the injured lung parenchyma of *Atm*-deficient mice as an indication of incipient tissue damage during A-T. Therefore, ASCs isolated from luciferase transgenic mice (mASCs) were intravenously transplanted into *Atm*-deficient and wild-type mice. Retention kinetics of the cells were monitored using in vivo bioluminescence imaging (BLI) and completed by subsequent verification using quantitative real-time polymerase chain reaction (qRT-PCR). The in vivo imaging and the qPCR results demonstrated migration accompanied by a significantly longer retention time of transplanted mASCs in the lung parenchyma of *Atm*-deficient mice compared to wild type mice. In conclusion, our study suggests incipient damage in the lung parenchyma of *Atm*-deficient mice. In addition, our data further demonstrate that a combination of luciferase-based PCR together with BLI is a pivotal tool for tracking mASCs after transplantation in models of inflammatory lung diseases such as A-T.

## 1. Introduction

Pulmonary failure is a frequent cause of morbidity and mortality in Ataxia-telangiectasia (A-T). At present, no effective pulmonary therapy for A-T exists [1]. Thus, the development of new strategies to preserve lung function in A-T is urgently needed due to limited clinical intervention options. Aside from immunodeficiency and inflammation, fibrotic changes are a major factor leading to progressive lung destruction. A direct connection between the ATM protein (A-T Mutated) and TGF-β_1_, one of the key mediators responsible for fibrotic changes in the lung, has been described [2]. In addition, we provided evidence for reduced lung function and increased inflammation in the lung of *Atm*-deficient mice displaying the human pulmonary A-T phenotype [3]. Therefore, inhibition of inflammation and fibrosis might open new avenues in the treatment of the lung disorder in A-T. Recently, we demonstrated that bone marrow transplantation (BMT) significantly improves the immunological phenotype and inhibits tumorigenesis in *Atm*-deficient mice [4]. Donor bone marrow derived cells (BMDCs) migrated into bone marrow, blood, thymus, spleen, and lung tissue of *Atm*-deficient mice. However, although the BMT overcame the immunodeficiency, migration of the donor cells into the lung tissue was low, and most of the cells were of hematopoietic origin. To improve cellular migration and to provide anti-inflammatory, anti-fibrotic, and anti-oxidative activity, a promising approach could be transplantation of mesenchymal stromal/stem cells (MSCs). 

In principle, MSCs have been detected throughout the body as immature, undifferentiated cells. For the first time, their isolation from bone marrow was described, but in the meantime, they have also been described from almost all adult tissues (e.g., fatty tissue) and solid organs (e.g., liver, kidney) [5,6]. More recent data show that MSCs represent a rare population (or populations) in the perivascular niche of all tissues. A number of studies have demonstrated that MSCs preferentially migrate to injured lung tissue where they are involved in lung repair and control of injury [7]. MSCs provide both structural and functional support to the parenchymal cells of multiple organs and possess immunomodulatory, anti-fibrotic properties and relative immune privilege [8]. MSCs release a number of anti-inflammatory, proangiogenic, regeneration-promoting, and immunomodulating factors that can improve regeneration in injured cells in tissues and organs [9]. Furthermore, MSCs preferentially migrate into injured or inflamed tissues. Therefore, we investigated in this study whether infused murine adipose-derived MSCs (mASCs) displayed an increased retention in injured pulmonary tissue of *Atm*-deficient mice compared to wild type mice, which could then result in an increased regeneration of the damaged lung tissue.

## 2. Materials and Methods 

### 2.1. Animals

*Atm*-deficient mice (*Atm*^tm1(Atm)Awb^; 8 to 10 weeks old) and corresponding wild-type mice, in a 129S6/SvEv background, were used as the animal model. A total of 24 *Atm*-deficient mice were included in the study (histological lung sections, *n* = 4; lung function testing, *n* = 12; bioluminescence imaging and PCR, *n* = 8). The experiments were performed using respective wildtype controls (littermates). Transgenic Luc^+^ mice with C57BL/6 background were used to isolate mesenchymal stromal/stem cells from murine inguinal fat. All animal procedures were performed according to the protocols approved by the Animal Care and Use Committee of the state of Hessen (RP Darmstadt (Gen. Nr. FK/1034)).

### 2.2. Cell Isolation and Culture 

Adipose tissue was harvested from transgenic Luc^+^ C57BL/6 mice (Janvier, France) as described earlier [10]. Briefly, mice were killed by cervical dislocation, and adipose tissue from inguinal fat pads was immediately dissected to isolate adipose-derived stromal/stem cells (mASCs). Tissue was minced with two scalpels (crossed blades) and then incubated in a 0.5% collagenase/phosphate buffered saline (PBS) solution (Collagenase Type: CLS; Biochrom, Berlin, Germany; PBS; Sigma, Taufkirchen, Germany) for 1 h at 37 °C with constant shaking. The digested tissue solution was then separated through a 100 µm strainer, and the resulting filtrate was centrifuged at 300× *g* for 5 min. The resulting pellet was washed twice with medium and centrifuged again at 300× *g* for 5 min. Finally, cells were plated and cultured at 37 °C in an atmosphere of 5% CO_2_ in 100% humidity. Dulbecco’s Modified Eagle’s Medium (DMEM; Sigma, Taufkirchen, Germany) with a physiological glucose concentration (100 mg/dL) was supplemented with 10% fetal bovine serum (Biochrom, Berlin, Germany) and used as standard culture medium (DF10). The medium was replaced every three days. Subconfluent cells (90%) were passaged by trypsinization. Cells between passages 2 and 5 were used throughout the experiments. Cell morphology was examined by phase contrast microscopy and flow cytometry, as described earlier [10].

### 2.3. Immunohistochemistry

Mice were anesthetized with an intraperitoneal injection of a Ketamin–Rompun mixture (20% Ketamin, CuraMED GmbH, Karlsruhe, Germany; 8% Rompun, Bayer Vital GmbH, Leverkusen, Germany). They were perfused transcardially with 4% paraformaldehyde in PBS. Lung tissue sections were prepared from fixed, paraffin-embedded organs and stained with hematoxylin/eosin or with chloracetate esterase staining as neutrophil-specific marker [4,11].

### 2.4. Pulmonary Function

Pulmonary function was tested in *Atm*-deficient mice and wild-type mice using a computer-controlled piston ventilator (flexiVent, SCIREQ Inc., Montreal, QC, Canada). Briefly, mice were anesthetized, a tracheotomy was performed, and the trachea was cannulated. After that, the mice were placed on a temperature controlled heat blanket, the trachea was exposed, and the previously calibrated cannula (1.2 cm, 18 gauges) was inserted and fixed using a suture. Ventilation was maintained at a rate of 150 breaths/min, a tidal volume of 10 mL/kg, and a positive end-expiratory pressure of 3 cm of water. Mice were allowed to acclimate to the ventilator for 2–3 min before measurement. Lung function parameters were calculated by fitting pressure and volume data to the single compartment model by measuring respiratory system resistance (Rrs), dynamic compliance (Crs) and elastance (Ers) and by analyzing with flexiWare 7 Software [3].

### 2.5. Transplantation 

Three wildtype mice and 4 *Atm*-deficient mice were transplanted with 0.5 × 10^5^ Luc^+^ mASCs in DMEM containing Heparin (10 U/100 µL). Viability of the cells was checked by trypan blue exclusion immediately before transplantation. A single injection of 100 µL was conducted via the tail vein into each mouse. Cells were tracked via bioluminescence imaging (BLI) on days 1, 3, 6, and 9 and via qRT-PCR at the indicated endpoints (day 15 and 50). The mice were weighed every two to three days, and no significant weight changes could be detected. The mice were sacrificed at the end of the experiment by cervical dislocation under anesthesia with ketamine-xylazine, and the organs were collected. 

### 2.6. In Vivo Bioimaging

Tracking of the transplanted Luc^+^ mASC was performed in vivo using the PerkinElmer IVIS Lumina II Imaging Chamber System. For this purpose, the mice were injected i.p. with 100 µL D-luciferin. They were then placed under anesthesia with isoflurane. After 10 min, the mice were placed dorsally or ventrally next to one another in the heated measuring chamber of the bioimaging system. Anesthesia was maintained during the measurement via a breathing mask. Images of the detectable luminescence signal were acquired at 1, 30, 60, 180, and 360 s and then evaluated using the “LivingImage” software with Region of Interest (ROI) placed over the thorax. In addition, a background field was placed on the dark border. The quantification was done in photons/s (total flux), and the background was subtracted from the measured values.

### 2.7. PCR Analyses

Total RNA extraction was performed using single-step RNA isolation from cultured cells by a standard protocol, as described earlier [10]. After RNA extraction, cDNAs were synthesized for 30 min at 37 °C using 1 µg of RNA, 50 µM random hexamers, 1 mM of deoxynucletide-triphosphate-mix, 50 units of reverse transcriptase (Fermentas, St. Leon-Rot, Germany) in 10× PCR buffer, 1 mM β-mercaptoethanol, and 5 mM MgCl_2_. An Absolute qPCR SYBR Green Rox Mix was used (Thermo Scientific, Hamburg, Germany) for the master mix; primer mix and RNAse-free water were added. Quantitative PCR was carried out in 96-well plates using the following conditions: 15 min at 95 °C for enzyme activation, 15 s at 95 °C for denaturation, 30 s for annealing, 30 s at 72 °C for elongation, followed by a melting curve analysis. Products were checked by agarose gel electrophoresis in selected experiments. Quantification of the PCR fragment was carried out using the Eppendorf realplex2 Mastercycler epgradient S (Eppendorf, Hamburg, Germany). Standard curves were prepared for the amplification specific efficiency correction, and the efficiencies (E) were calculated according to the equation E = (10^−1^/m^−1^) × 100, where m is the slope of the linear regression model fitted over log-transformed data of the input cDNA concentrations versus CT values [12,13]. E for actin-beta was 1.9164, and for luciferase = 1.9399. The relative efficiency-corrected mRNA expression of the target gene was calculated based on efficiencies and the CT (Threshold cycle) deviation of an unknown sample versus a positive control (ΔCT) and was expressed in comparison to a reference gene [12,13]. Data were calculated (rel. expression = (E Luc = 1.9164)^ΔCT Luc^/(E actin = 1.9399)^ΔCT actin^) and expressed as percent (the calculated rel. expression 1.0 refers to 100%, the value 0.02 refers to 2%). The luciferase primer was constructed using the firefly luciferase gene from *Photinus pyralis* (GenBank No. AB644228.1, forward: TGAAGAGATACGCCCTGGTT, reverse: CTACGGTAGGCTGCGAAATG; product size 288 bp) and the reference primer for *murine actin-beta* (NM_007393, forward: F: CCACCATGTACCCAGGCATT, reverse: AGGGTGTAAAACGCAGCTCA, product size 253 bp) (Invitrogen (Karlsruhe, Germany). In addition, PCR products were separated by agarose electrophoresis and observed under UV illumination [10]. 

### 2.8. Statistical Analysis 

The data were expressed as mean ± standard deviation (SD) and were analyzed using an unpaired student’s Test or a Mann–Whitney test. For multiple comparisons, analysis of variance (ANOVA) with Bonferroni’s multiple comparison test was used for statistical analysis. *p* values < 0.05 were considered significant.

## 3. Results

### 3.1. Atm-Deficient Mice Exhibited Signs of Lung Disease

Comparison of the lung parenchyma showed slight tendency for alveolar septal thickening and patchy areas of neutrophilic inflammation in *Atm*-deficient mice (Figure 1B,D) compared to wild-type mice (Figure 1A,C) accompanied by significantly increased lung resistance (Rrs) (Figure 1E) and respiratory system elastance (Ers) (Figure 1F) as well as decreased tissue compliance (Figure 1F) in comparison to control mice.

### 3.2. Luc^+^ mASCs Stayed Longer in Lung Tissue of Atm-Deficient Mice Compared to Wildtype Mice

Examination of in vivo luciferase expression showed a positive bioluminescent signal in all transplanted mice on day one after transplantation with no differences between *Atm*-deficient and wildtype mice (Figure 2A). After three days, the bioluminescent signal in wildtype mice rapidly dropped down and disappeared completely on day six. In contrast, the transplanted *Atm*-deficient mice showed a strong bioluminescent signal even after nine days. At day 14, the bioluminescence signal decreased to undetectable levels in all mice including the four *Atm*-deficient mice (data not shown).

Quantification of the light emission of the analyzed bioluminescence signals confirmed the above findings (Figure 2B). A strong signal was seen on day one for both genotypes after transplantation, whereas the mock-transplanted animals, which were signal negative in the overlay recordings, showed a constant-low total flux background value on all days. Up to day three, the total flux of the bioluminescent signal revealed no differences between MSC-transplanted *Atm*-deficient and wildtype mice. After that, the signal in transplanted wildtype mice dropped down to the level of the untransplanted mice. In contrast, the total flux signal maintained in the MSC-transplanted *Atm*-deficient mice.

### 3.3. Transplanted mASCs Exhibited a Long Retention Time in the Lung Parenchyma of Atm-Deficient Mice

Quantitative real-time PCR was used on days 15 and 50 to follow the retention time of Luc^+^ mASCs after the bioluminescence signal disappeared (Figure 3). After transplantation of Luc^+^ mASCs, *Atm*-deficient mice showed a 50-fold higher luciferase gene expression in lung parenchyma compared to wildtype mice (*Atm*^−/−^: 2.0% ± 0.59; *Atm*^+/+^: 0.04% ± 0.004, *p* < 0.05), whereas no differences could be detected in the kidney and the thymus (Figure 3). 

An exemplary examination of long-term stay revealed still a positive Luc-signal in the lung parenchyma of the *Atm*-deficient mouse at day 50 (*n* = 1). Although the Luc signal in the lungs decreased on day 50 compared to day 15, it was still detectable (Figure 4). While a very slight increase of the signal was observed in the kidney at day 50, no signal was detected in the thymus on day 15 or day 50.

## 4. Discussion

Respiratory disease accounts for significant morbidity and mortality in patients with A-T [14]. Major phenotypes include recurrent respiratory tract infections and bronchiectasis, aspiration, respiratory muscle abnormalities, interstitial lung disease, and pulmonary fibrosis [15]. Aside from the immunodeficiency, it has been proposed that ongoing low grade inflammation and oxidative stress might be responsible for the clinical pathogenesis causing lung failure [1,16]. Recent studies further demonstrated restricted lung function, high sensitivity to inflammatory agents, and a significant amount of oxidative DNA damage in the lung parenchyma of *Atm*-deficient mice, especially after triggering inflammation [3,11].

Cell therapy using MSCs might be a promising approach for tissue regeneration in A-T. MSCs have been shown to integrate into the damaged sites in a variety of tissues, including lung tissue, showing positive effects on tissue regeneration [7]. There are recent studies using administration of MSCs or their derivates (e.g., extracellular vesicles) in in vivo studies [17] or human clinical trials to treat pulmonary fibrosis [18]. Averyanov and co-workers evaluated the safety and the tolerability of repeated infusions of high doses of MSCs up to the total cumulative dose of two billion cells in subjects with rapidly progressing idiopathic pulmonary fibrosis. They showed that the treatment was safe and well tolerated. Transplanted patients had an increased lung function compared to the placebo group, where a sustained decrease in lung function was observed. Currently, it could be shown that MSCs improved the outcome of patients with COVID-19 pneumonia [19]. Transplantation of MSCs was shown to cure or significantly improve the functional outcomes of seven COVID-19 patients without observed adverse effects. The pulmonary function and symptoms of seven patients were significantly improved in two days after MSC transplantation [19]. Nevertheless, it should be noted in this context that others reported that reduced migration of transplanted MSCs correlated with decreased fibrosis in the lungs [20].

However, lung disease in A-T is a creeping process that slowly develops over time, and experience with MSCs in A-T is scarce. Even in the *Atm*-mouse model, lung damage is difficult to determine without induction of experimental inflammation, as the mice died from thymic lymphomas within 3–6 months before lung failure could occur. This prompted us to investigate whether mASCs remained in the lung parenchyma of *Atm*-deficient mice in an increased manner compared to healthy controls. It is important to note that the mice were housed in individually ventilated cages to protect them from any harmful microorganism.

Although tissue damage and inflammation are hard to detect in *Atm*-deficient mice without exogenous trigger, our present data showed signs of lung disease and damage in *Atm*-deficient mice [3,4]. These findings are underlined by our in vivo imaging and qPCR results, which demonstrated increased retention of MSCs into the lung parenchyma of *Atm*-deficient mice compared to wild-type. Because MSCs preferentially migrate into injured or inflamed tissues, such as during wound healing or in association with tumors, these findings together with the pathological changes in lung function parameters indicate that the *Atm*-deficient mouse has some kind of ongoing pulmonary damage [21]. It should be mentioned that signals in the lung do not necessarily reflect transgenic donor cells. They may derive from inflammatory host cells (e.g. macrophages or others) in the lung, which have phagocytosed donor cells. Nevertheless, our results shown here are in line with earlier findings showing an engraftment of CD31^+^CD45^−^ endothelial cells and EpCAM+ epithelial cells into the lung tissue of *Atm*-deficient mice after bone marrow transplantation [4]. In addition, impaired lung resistance and respiratory system elastance in *Atm*-deficient mice has also been described in other studies without the presence of inflammation [3,11]. In this regard, an earlier study from our group also showed increased spontaneous oxidative stress and damage in the lung tissue of *Atm*-deficient mice and alveolar basal epithelial cells in the presence of the ATM-kinase inhibitor KU55933 [3,16]. Thus, in the absence of inflammatory signals, oxidative stress could attract MSCs in our experimental setting. Due to their ability to counteract reactive oxygen species, further experiments investigating the effect of MSCs on oxidative stress should be performed. Luc-based real-time PCR together with BLI is an important tool for cell tracking after transplantation in models of inflammatory lung diseases such as A-T [10].

In conclusion, tracking of transplanted Luc^+^ mASCs, by a combination of luciferase-based PCR together with BLI showed an increased migration into lung parenchyma accompanied by a significantly longer retention time in *Atm*-deficient mice, pointing to ongoing pulmonary damage in the lung tissue in these animals. To what extent these cells improve the regeneration of damaged lung tissue in *Atm*^−/−^ mice will now be investigated in further studies. Therefore, further experiments are necessary to confirm the regenerative impact of MSCs on lung disease in A-T.

## Figures and Tables

**Figure 1 cells-09-01444-f001:**
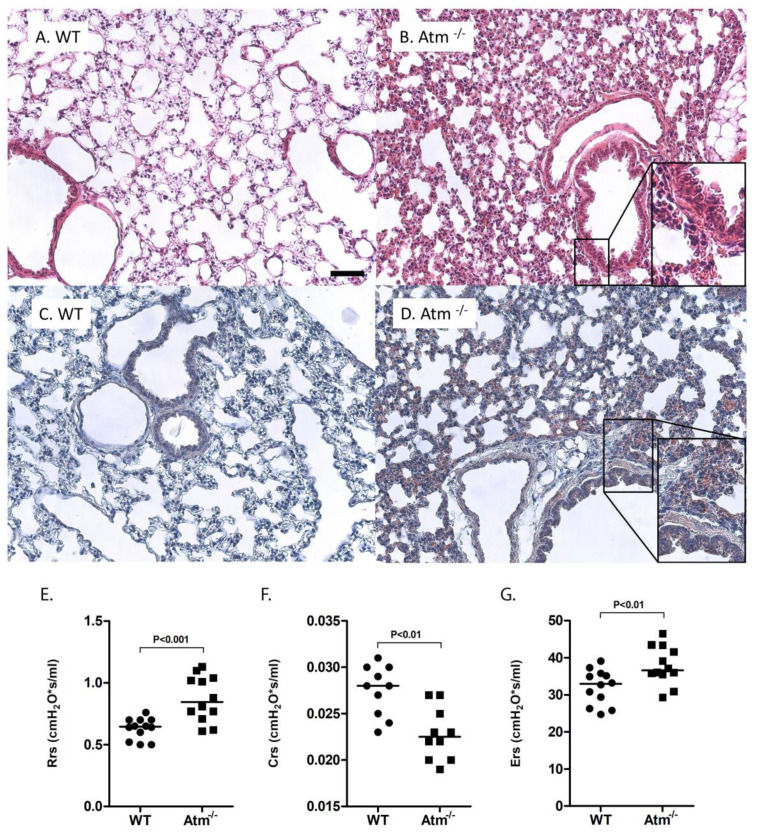
Lung injury in *Atm*-deficient mice. Representative histological lung sections of (**A**,**C**) wild-type (WT) and (**B**,**D**) *Atm*-deficient (Atm^−/−^) mice stained with hematoxylin and eosin (**A**,**B**) or with chloracetate esterase staining (**C**,**D**), respectively. Lung function testing using a FlexiVent mouse ventilator, respiratory system resistance (**E**), compliance (**F**), and elastance (**G**) in *Atm*^−/−^ mice compared to a WT control group (*n* = 10–12). Bar = 100 µm.

**Figure 2 cells-09-01444-f002:**
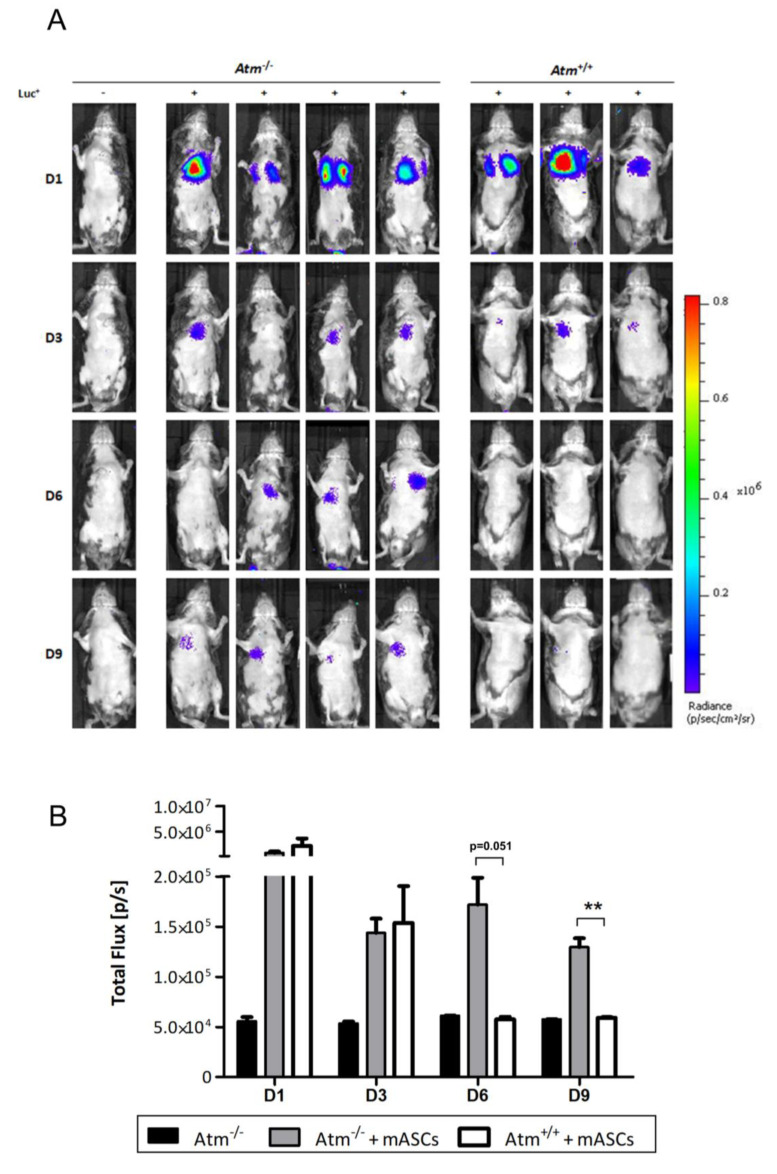
Retention of adipose-derived mesenchymal stromal/stem cells (ASCs) isolated from luciferase transgenic mice (mASCs) into the lung tissue of *Atm*-deficient mice. (**A**) Bioluminescence imaging of mASCs transgenic for the firefly luciferase gene (Luc^+^) in the lung tissue of *Atm*-deficient mice (*Atm*^−/−^, *n* = 4) and wild type mice (*Atm*^+/+^, *n* = 3). MSCs were analyzed at days 1, 3, 6, and 9 after transplantation. (**B**) Quantitative analysis of the light emission data of the analyzed bioluminescence signals from untransplanted *Atm*^−/−^ mice (black bars, *n* = 3), Luc^+^ mASCs transplanted *Atm*^−/−^ mice (grey bars, *n* = 4), and *Atm*^+/+^ mice (white bars, n=3). ** *p* < 0.01.

**Figure 3 cells-09-01444-f003:**
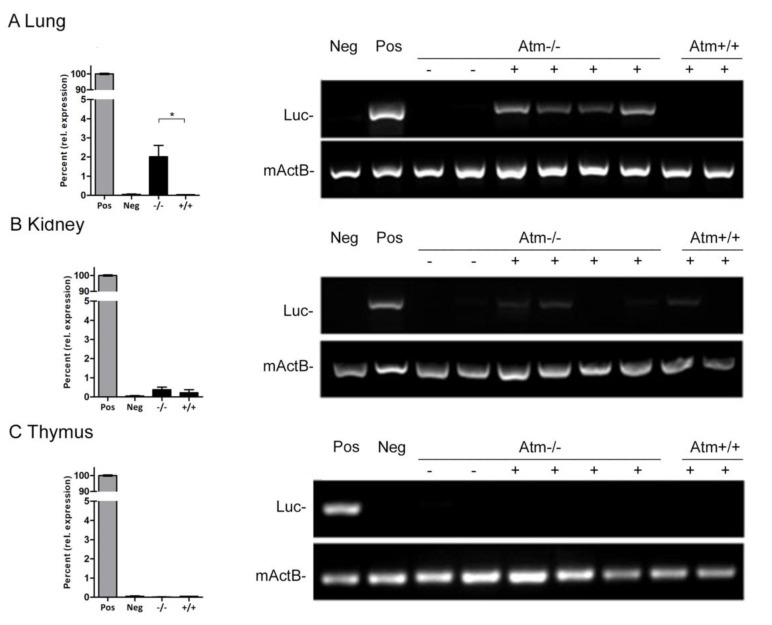
Detection of Luc^+^ mASCs on day 15 after transplantation of *Atm*-deficient (*Atm*^−/−^, *n* = 4) and wild type mice (*Atm*^+/+^, *n* = 3) knock-out using PCR. Quantitative analysis of the relative expression (expressed as percent; MW ± SEM) (left side) and results of the gel electrophoresis of lungs, kidney, and thymus (right side). Controls: RNA from Luc negative tissue (Neg), RNA from Luc positive tissue (Pos = 100%). Luc=luciferase, mActB= murine actine-beta. * *p* < 0.05.

**Figure 4 cells-09-01444-f004:**
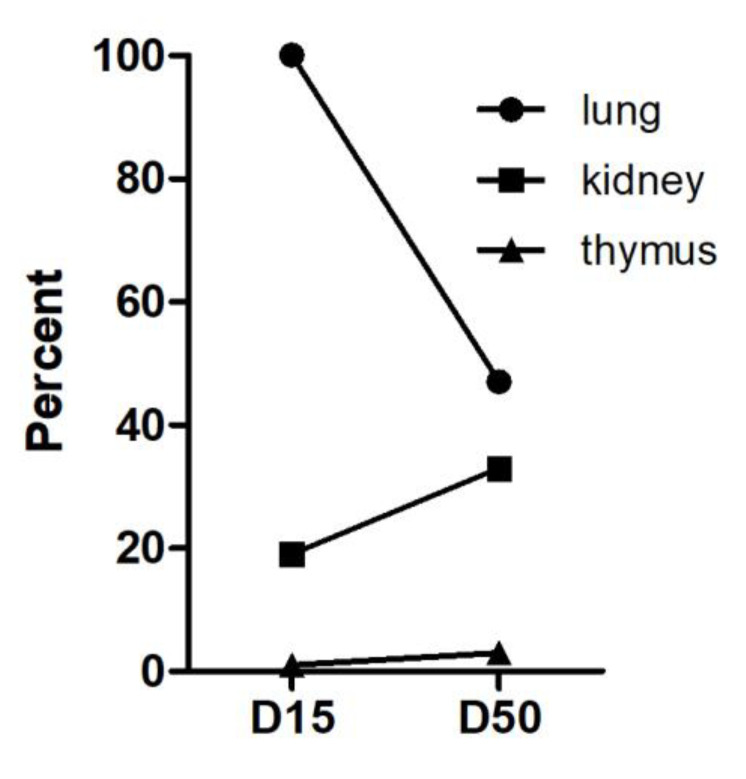
Detection of Luc^+^ mASCs on day 15 (D15, *n* = 4) and on day 50 (D50, *n* = 1) after transplantation in *Atm*^−/−^ mice. Graph shows fold Luc^+^ expression in lung, kidney, and thymus at days 15 and 50 in relation to Luc+ expression in the lung at day 15 (quantitative analysis of the relative Luc-mRNA amounts (percent of signal, lung D15 = 100%)).

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
