# Peer review of "Tracking of Infused Mesenchymal Stem Cells in Injured Pulmonary Tissue in Atm-Deficient Mice"

_cells, 2020, doi:10.3390/cells9061444_

Round 1
Reviewer 1 Report
In the article by Baer et al., the authors have investigated the recruitment of murine adipose-derived MSCs (mASCs) to the lungs of Atm-deficient mice. Recruitment was investigated by In vivo bioimaging and qPCR. Although an interesting premise, the authors have failed to capitalise on their experiments and consequently the manuscript would benefit from several additional experiments.
From MM unclear of the group sizes, and how experiments were performed. One experiment with all mice, or several experiments with smaller group numbers. Please clarify in the manuscript. The authors also state they weighed every two days, please include or discuss this data.
Please include scale bars in the Fig1A, and high quality magnified areas showing the different lung compartments; airways, vessels and parenchyma. To help the reader understand the remodelling changes that are induced by Atm deficiency, please also include additional Masson’s Trichome staining, to highlight collagen deposition. Have the authors quantified the remodelling found in the Atm-deficient mice?
Which inflammatory cells are recruited to the lungs of Atm deficient mice?
In the lung function experiments, was PV loops performed? If we please include the associated derived parameters e.g. quasi-static lung compliance (Cst), A or Area. This is especially important as Rrs, Crs and Ers derived from the Snapshot, are restricted to changes in the upper airway.
Here I commend the authors for presenting individual mice in the associated graphs.
In Figure 3, please change the data presentation from rel. mRNA expression (which is not described in the Material and Methods) to show either ΔCT values or ΔΔCT. Unfortunately, 2-ΔΔCT nor rel. mRNA expression helps data presentation or interpretation.
In Figure 4, is there any statistical significance between any of the groups? If not what is the interpretation of this figure.
Please show either immunohistochemical or immunofluorescent staining against Luc+ mASC in the lungs at all time points, at a minimum D3, 9, 15 and 50 must be shown. Where do the mASC localise? Please use the appropriate co-staining to confirm this localisation.
Please explain why are mASC lost quicker from WT mice compared to atm-deficient mice?
Please include lung function measurements after mASC transfer (e.g. D3, 9, 15 and 50); did the mASC produce any functional improvement? (Similarly the authors could also include lung histology or quantification of inflammatory cells)
Minor
Although not strictly necessary, it is more usual to give the WT (control) groups first in the graphs (Figure 1).
Typo: Although, tissue damage and inflammation is hardly to detect in Atm-deficient mice... Hardly, should be hard.
Author Response
Dear Reviewer #1,
Thank you for your kind and very useful comments. We have considered all comments carefully and revised the manuscript in accordance with your suggestions whenever it was possible.
In the article by Baer et al., the authors have investigated the recruitment of murine adipose-derived MSCs (mASCs) to the lungs of Atm-deficient mice. Recruitment was investigated by In vivo bioimaging and qPCR. Although an interesting premise, the authors have failed to capitalise on their experiments and consequently the manuscript would benefit from several additional experiments.
- From MM unclear of the group sizes, and how experiments were performed. One experiment with all mice, or several experiments with smaller group numbers. Please clarify in the manuscript. The authors also state they weighed every two days, please include or discuss this data.
Thank you for your comment. In addition to the figure legends, we added group sizes, and more detailed performance of the experiments to the M&M section.
A total of 24 Atm-deficient mice were included in the study (histological lung sections, n=4; lung function testing, n=12; bioluminescence imaging and PCR, n=8). The experiments were carried out using respective wildtype controls (littermates).
We also included the outcome of the weight measures to the manuscript. Because transplantation did not significantly influenced the weight of the animals we decided not to include the data into the manuscript.
- Please include scale bars in the Fig1A, and high quality magnified areas showing the different lung compartments; airways, vessels and parenchyma. To help the reader understand the remodelling changes that are induced by Atm deficiency, please also include additional Masson’s Trichome staining, to highlight collagen deposition. Have the authors quantified the remodelling found in the Atm-deficient mice? Which inflammatory cells are recruited to the lungs of Atm deficient mice?
Thank you for your very valuable and useful comments. We improved the magnification and quality of the figures to make changes in the lung architecture more visible to the reader (and also added a scale bar). Unfortunately, we did not stain with Masson’s Trichome when we performed the experiments, but we added chloracetate esterase staining to show neutrophilic inflammation in Atm-deficient mice (Fig. 1D, versus wildtypes, Fig. 1D).
- In the lung function experiments, was PV loops performed? If we please include the associated derived parameters e.g. quasi-static lung compliance (Cst), A or Area. This is especially important as Rrs, Crs and Ers derived from the Snapshot, are restricted to changes in the upper airway.
Here I commend the authors for presenting individual mice in the associated graphs.
We are aware that the reviewers’ suggestion might be a more powerful method to evaluate respiratory disease in mice. However, assessing lung function parameters by single frequency Forced Oscillation Technique (FOT) measurements still provide valuable information on the respiratory system. As evaluated by Vanoirbeek et al, this “snapshot perturbation” maneuver is able to measure resistance (R), compliance (C), and elastance (E) of the whole respiratory system (airways, lung, and chest wall) [Vanoirbeek et al.].
As shown before, there are several factors which influence the pulmonary disease in A-T including inflammation, oxidative stress and impaired damage repair processes that occur in the lung over the time and our results are in line with earlier investigations of the respiratory system in Atm-deficient mice (McGrath-Morrow SA, Pediatr Pulmonol. 2010, Eickmeier O, BMC Pulm Med 2014).
We thank the reviewer for the suggestion and we will definitely keep this in mind when we do further lung function experiments.
- Vanoirbeek JA, Rinaldi M, De Vooght V, Haenen S, Bobic S, Gayan-Ramirez G, Hoet PH, Verbeken E, Decramer M, Nemery B, Janssens W. Noninvasive and invasive pulmonary function in mouse models of obstructive and restrictive respiratory diseases. Am J Respir Cell Mol Biol. 2010;42:96–104.
- McGrath-Morrow SA, Gower WA, Rothblum-Oviatt C, Brody AS, Langston C, Fan LL, Lefton-Greif MA, Crawford TO, Troche M, Sandlund JT, Auwaerter PG, Easley B, Loughlin GM, Carroll JL, Lederman HM. Evaluation and management of pulmonary disease in ataxia-telangiectasia. Pediatr Pulmonol. 2010 Sep;45(9):847-59.
- Olaf Eickmeier, Su Youn Kim, Eva Herrmann, Constanze Döring, Ruth Duecker, Sandra Voss, Sibylle Wehner, Christoph Hölscher, Julia Pietzner, Stefan Zielen, Ralf Schubert. Altered Mucosal Immune Response After Acute Lung Injury in a Murine Model of Ataxia Telangiectasia. BMC Pulm Med. 2014 May 29;14:93.
- In Figure 3, please change the data presentation from rel. mRNA expression (which is not described in the Material and Methods) to show either ΔCT values or ΔΔCT. Unfortunately, 2-ΔΔCT nor rel. mRNA expression helps data presentation or interpretation.
We changed the data presentation from rel. mRNA expression to percent (expression 2-ΔΔCT), as described in the M&M section, accordingly.
- In Figure 4, is there any statistical significance between any of the groups? If not what is the interpretation of this figure.
As described in the manuscript “An exemplary examination of long term stay revealed still a positive Luc-signal in the lung parenchyma of the Atm-deficient mouse at day 50. This was only one experiment, to look if we still can detect a positive signal at day 50. But we feel this is an important information for the readers. However, to avoid redundancy, we changed the figure showing fold luc expression in lung, kidney and thymus at day 50 in relation to lung at day 15 (=100%). If the reviewer does not find this procedure to be good, we would remove this figure from the publication.
- Please show either immunohistochemical or immunofluorescent staining against Luc+ mASC in the lungs at all time points, at a minimum D3, 9, 15 and 50 must be shown. Where do the mASC localise? Please use the appropriate co-staining to confirm this localisation.
We totally agree with the reviewer that further immunohistochemical and immunofluorescent data at D3, 9, 15 and 50 has to be shown to proof the effect of mASC on lung injury and damage in Atm-deficient mice. Indeed, this is what we are going do to in an further project supported by a DGF-grant. There we will answer the question whether MSCs have an effect on inflammation in A-T. However, the message of the present brief report / short communication is that tracking of transplanted Luc+ mASCs, by a combination of luciferase-based PCR together with BLI, showed an increased migration into lung parenchyma accompanied by a significant longer retention time in Atm-deficient mice. To what extent these cells improve the regeneration of damaged lung tissue in Atm-deficient mice will now be investigated in further studies.
- Please explain why are mASC lost quicker from WT mice compared to atm-deficient mice?
This is an interesting question. MSCs preferentially migrate to injured lung tissue where they are involved in lung repair and control of injury [Huleihe 2013]. For our knowledge MSC in WT mice did not migrate into the lung parenchyma because there are no signs of inflammation or fibrosis. Indeed, Epperly et al. have demonstrated that reduced migration of transplanted MSCs correlates with decreased fibrosis into the lung parenchyma in a Smad3-/- chimeric mouse model [Epperly 2006]. We added this to the discussion section.
Huleihel L, Levine M, Rojas M. The potential of cell-based therapy in lung diseases. Expert Opin Biol Ther. 2013 Oct;13(10):1429-40.
Epperly, M.W.; Franicola, D.; Zhang, X.; Nie, S.; Wang, H.; Bahnson, A.B.; Shields, D.S.; Goff, J.P.; Shen, H.; Greenberger, J.S. Reduced irradiation pulmonary fibrosis and stromal cell migration in Smad3-/- marrow chimeric mice. In Vivo 2006, 20, 573–582.
Please include lung function measurements after mASC transfer (e.g. D3, 9, 15 and 50); did the mASC produce any functional improvement? (Similarly the authors could also include lung histology or quantification of inflammatory cells)
Please see above (point 6)
Minor
Although not strictly necessary, it is more usual to give the WT (control) groups first in the graphs (Figure 1).
We followed the reviewers’ advice and give the WT (control) groups first in the graphs
Typo: Although, tissue damage and inflammation is hardly to detect in Atm-deficient mice... Hardly, should be hard.
We changed hardly to hard.
Reviewer 2 Report
General Comments
This is a concise study that evaluates the dynamics of ASC engraftment into a murine model of pulmonary fibrosis. The findings are timely in light of increased attention being given to the COVID-19 pandemic and its clinical course. The authors have presented complementary findings validating their conclusions. Some aspects of the work could be improved as outlined below.
- Not all studies are consistent with the conclusions of the current manuscript (see ref below). Others have reported that reduced intra-pulmonary migration of BMSC correlated with reduced fibrosis. This aspect of the literature needs to be incorporated into the discussion
Epperly MW et alReduced irradiation pulmonary fibrosis and stromal cell migration in Smad3-/- marrow chimeric mice. In Vivo 2006 Sep-Oct;20(5):573-82
- There are recent studies using MSC transplant in human clinical trials to treat pulmonary fibrosis that should be incorporated into the discussion. Examples are below:
Averyanov A et al. First-in-human high-cumulative-dose stem cell therapy in idiopathic pulmonary fibrosis with rapid lung function decline. Stem Cell Translational Medicine 2020 Jan;9(1):6-16.
- There are mechanism based papers using MSC derived exosomes to consider incorporating into the discussion.
Mansouri N et al Mesenchymal stromal cell exosomes prevent and revert experimental pulmonary fibrosis through modulation of monocyte phenotypes. JCI Insights 2019 Nov 1;4(21). pii: 128060. doi: 10.1172/jci.insight.128060.
- The recent description of MSC transplant to treat COVID-19 patients provides a rationale for increased attention to the dynamics of MSC engraftment within pulmonary tissue. This application of the current findings should be highlighted in the concluding statement of the Discussion.
Leng Z et al. Transplantation of ACE2- Mesenchymal Stem Cells Improves the Outcome of Patients with COVID-19 Pneumonia. Aging Dis 2020 Mar 9;11(2):216-228. doi: 10.14336/AD.2020.0228. eCollection 2020 Apr.
- The number of replicate animals used in the study is small but proved sufficient to obtain statistically significant outcomes at Day 15 time points; however, no standard deviation is displayed for Day 50 outcomes. Some comment regarding the cohort size at the final time point is warranted.
Specific Comments
Methods, 2.4. Was the tracheotomy procedure a survival surgery on the individual mice? Or was this a terminal event?
Figure 3 displays RT-PCR gels stained with ethidium bromide to demonstrate gene expression levels; however, the authors describe using delta/delta CT values in their methods, implying the use of qRT-PCR. Please clarify this apparent discrepancy.
Figure 4. The figure legend does not indicate how many animals were used in each time point. While the methods report that 3 ATM+/+ and 4 ATM-/- mice were examined at day 15, no information is described for the animals at day 50. Please clarify this matter in the figure legend and, if possible, explicitly describe the day 50 time point in the methods section.
Author Response
Dear Reviewer #2,
Thank you for your kind and very useful comments. We have considered all comments carefully and revised the manuscript in accordance with your suggestions whenever it was possible.
General Comments
This is a concise study that evaluates the dynamics of ASC engraftment into a murine model of pulmonary fibrosis. The findings are timely in light of increased attention being given to the COVID-19 pandemic and its clinical course. The authors have presented complementary findings validating their conclusions. Some aspects of the work could be improved as outlined below.
- Not all studies are consistent with the conclusions of the current manuscript (see ref below). Others have reported that reduced intra-pulmonary migration of BMSC correlated with reduced fibrosis. This aspect of the literature needs to be incorporated into the discussion
Epperly MW et alReduced irradiation pulmonary fibrosis and stromal cell migration in Smad3-/- marrow chimeric mice. In Vivo 2006 Sep-Oct;20(5):573-82
- There are recent studies using MSC transplant in human clinical trials to treat pulmonary fibrosis that should be incorporated into the discussion. Examples are below:
Averyanov A et al. First-in-human high-cumulative-dose stem cell therapy in idiopathic pulmonary fibrosis with rapid lung function decline. Stem Cell Translational Medicine 2020 Jan;9(1):6-16.
- There are mechanism based papers using MSC derived exosomes to consider incorporating into the discussion.
Mansouri N et al Mesenchymal stromal cell exosomes prevent and revert experimental pulmonary fibrosis through modulation of monocyte phenotypes. JCI Insights 2019 Nov 1;4(21). pii: 128060. doi: 10.1172/jci.insight.128060 .
- The recent description of MSC transplant to treat COVID-19 patients provides a rationale for increased attention to the dynamics of MSC engraftment within pulmonary tissue. This application of the current findings should be highlighted in the concluding statement of the Discussion.
Leng Z et al. Transplantation of ACE2- Mesenchymal Stem Cells Improves the Outcome of Patients with COVID-19 Pneumonia. Aging Dis 2020 Mar 9;11(2):216-228. doi: 10.14336/AD.2020.0228 . eCollection 2020 Apr.
We thank the reviewer for his valuable comments. We think that these suggestions will improve the discussion and adopt it. We added new sentences and the new literature to the discussion section:
“Cell therapy using MSCs might be a promising approach for tissue regeneration in A-T. MSCs have been shown to integrate into the damaged sites in a variety of tissues, including lung tissue, showing positive effects on tissue regeneration [7]. There are recent studies using administration of MSCs or their derivates (e.g. extracellular vesicles) in in vivo studies [16] or human clinical trials to treat pulmonary fibrosis [17]. Averyanov and co-workers evaluated the safety and tolerability of repeated infusions of high doses of MSCs up to the total cumulative dose of 2 billion cells in subjects with rapidly progressing idiopathic pulmonary fibrosis. They showed that the treatment was safe and well tolerated. Transplanted patients had an increased lung function compared to the placebo group, where a sustained decrease in lung function was observed. Currently, it could be shown that MSCs improved the outcome of patients with COVID-19 pneumonia [18]. Transplantation of MSCs was shown to cure or significantly improve the functional outcomes of seven COVID-19 patients without observed adverse effects. The pulmonary function and symptoms of seven patients were significantly improved in 2 days after MSC transplantation [18]. Nevertheless, it should be noted in this context that others reported that reduced migration of transplanted MSCs correlated with decreased fibrosis in the lungs [19]. “
- The number of replicate animals used in the study is small but proved sufficient to obtain statistically significant outcomes at Day 15 time points; however, no standard deviation is displayed for Day 50 outcomes. Some comment regarding the cohort size at the final time point is warranted.
Thank you for your comment. In addition to the figure legends, we added group sizes, and more detailed performance of the experiments to the M&M section.
Unfortunately we currently only have one mouse at day 50, but think that we should still show the result here in the study, because at this point luc + cells were still detectable. If the reviewer does not find this procedure to be good, we would remove this figure from the publication. This is only a short communication / brief report, we work on further results.
Specific Comments
Methods, 2.4. Was the tracheotomy procedure a survival surgery on the individual mice? Or was this a terminal event?
This was a terminal event.
Figure 3 displays RT-PCR gels stained with ethidium bromide to demonstrate gene expression levels; however, the authors describe using delta/delta CT values in their methods, implying the use of qRT-PCR. Please clarify this apparent discrepancy.
We have described the standard PCR with agarose gel including an appropriate literature in the method section. “In addition, PCR products were separated by agarose electrophoresis and observed under UV illumination [10].”
Figure 4. The figure legend does not indicate how many animals were used in each time point. While the methods report that 3 ATM+/+ and 4 ATM-/- mice were examined at day 15, no information is described for the animals at day 50. Please clarify this matter in the figure legend and, if possible, explicitly describe the day 50 time point in the methods section.
We think this point was answered in the answer to point 5 above.
Reviewer 3 Report
The Authors aimed to investigate whether murine adipose-derived MSCs (mASCs) migrate into the injured lung parenchyma of Atm-deficient mice.The Authors use in vivo bioluminescence imaging (BLI) to monitor the Retention kinetics of the cells and quantitative real-time polymerase chain reaction (qRT-PCR) to complete verification.
They demonstrated, that migration was accompanied by a significant longer retention time of transplanted mASCs in the lung parenchyma of Atm-deficient mice compared to wild type mice. Moreover they demonstrated also that a combination of luciferase-based PCR, together with BLI, is a pivotal tool for tracking mASCs after transplantation in models of inflammatory lung diseases such as A-T.
Major points could be improved in this Manuscript:
-The Authors sustained that "Comparison of the lung parenchyma showed slight tendency for infiltration of inflammatory cells and alveolar septal thickening in Atm-deficient mice (Fig. 1B) compared to wild-type mice (Fig. 1A)”. To sustain this sentence, better investigations should be performed. Have they performed inflammatory cell count? Macrophages, neutrophils and lymphocytes can be evaluated. In addition, specific stain for inflammatory cells are required instead E.E. stain. Please,improve this point.
-Figure 1: Please, insert pictures Magnification
-What does inflammatory infiltration look like after a few days of infusion in animal groups? Why didn't the Authors investigate lung structural changes few day after cells infusion? It is not only necessary to evaluate whether ASC cells remain in the tissue but it is equally important to evaluate their short and long term effects. Please discuss this aspect.
Author Response
Dear Reviewer #3,
Thank you for your kind and very useful comments. We have considered all comments carefully and revised the manuscript in accordance with your suggestions whenever it was possible.
The Authors aimed to investigate whether murine adipose-derived MSCs (mASCs) migrate into the injured lung parenchyma of Atm-deficient mice.The Authors use in vivo bioluminescence imaging (BLI) to monitor the Retention kinetics of the cells and quantitative real-time polymerase chain reaction (qRT-PCR) to complete verification.
They demonstrated, that migration was accompanied by a significant longer retention time of transplanted mASCs in the lung parenchyma of Atm-deficient mice compared to wild type mice. Moreover they demonstrated also that a combination of luciferase-based PCR, together with BLI, is a pivotal tool for tracking mASCs after transplantation in models of inflammatory lung diseases such as A-T.
Major points could be improved in this Manuscript:
- The Authors sustained that "Comparison of the lung parenchyma showed slight tendency for infiltration of inflammatory cells and alveolar septal thickening in Atm-deficient mice (Fig. 1B) compared to wild-type mice (Fig. 1A)”. To sustain this sentence, better investigations should be performed. Have they performed inflammatory cell count? Macrophages, neutrophils and lymphocytes can be evaluated. In addition, specific stain for inflammatory cells are required instead E.E. stain. Please,improve this point.
-Figure 1: Please, insert pictures Magnification
Thank you for this very valuable comment. We added chloracetate esterase staining to show neutrophilic inflammation in Atm-deficient mice (Fig. 1D) and compared this with wildtype mice (Fig. 1C). We also improved the magnification and quality of the figures to make changes in the lung architecture more visible to the reader.
- What does inflammatory infiltration look like after a few days of infusion in animal groups? Why didn't the Authors investigate lung structural changes few day after cells infusion? It is not only necessary to evaluate whether ASC cells remain in the tissue but it is equally important to evaluate their short and long term effects. Please discuss this aspect.
This is absolutely correct and needs to be investigated. As this is only a short communication / brief report, we currently work on further results. In these studies we also investigate the structural changes after MSCs transplantation and the long term effects.
We added a sentence in the conclusion: “To what extent these cells improve the regeneration of damaged lung tissue in atm mice will now be investigated in further studies.”
Round 2
Reviewer 1 Report
The authors have made some efforts to address my previous concerns, however, I still have two open points that require addressing.
Thank you for including chloracetate esterase staining to show neutrophilic inflammation in Atm-deficient mice (Fig. 1D, versus wildtypes, Fig. 1D), unfortunately, the included images are low quality (which may have happened during the conversion process) and appear to show positive staining throughout the lung parenchyma, in addition to the patchy areas adjacent to the airways. Based on the supplied image, it is not possible for me to determine whether this is due to non-specific staining or that all thickened septae saturated with neutrophils. Please improve the quality of supplied images.
The presentation of the real-time data in Figure 3 is still confusing for me. To help the reader please expand the material and methods to explain exactly how the “percent of expression 2-ΔΔCT” was calculated. i.e. the following statement is not sufficient “Relative quantification was estimated by the ddCT method [12] with b-actin as a calibrator. The level of target gene expression was calculated using 2-ddCT”. Please state exactly how dCT, ddCT, 2-ΔΔCT and ultimately percent of expression 2-ΔΔCT was calculated. In the analysis of this figure were only -/- and +/+ groups compared? And was “ANOVA with Bonferronni`s Multiple Comparisation Test” used for this comparison as stated in the material and methods section?
Thank you for clarifying that the day 50 timepoint (Figure 4) is from only one mouse.
Author Response
POINT TO POINT ANSWERS TO REVIEWER 1
-Dear Reviewer #1,
-Thank you for your kind and very useful comments. We added the revisions in the manuscript in blue.
The authors have made some efforts to address my previous concerns, however, I still have two open points that require addressing.
Thank you for including chloracetate esterase staining to show neutrophilic inflammation in Atm-deficient mice (Fig. 1D, versus wildtypes, Fig. 1D), unfortunately, the included images are low quality (which may have happened during the conversion process) and appear to show positive staining throughout the lung parenchyma, in addition to the patchy areas adjacent to the airways. Based on the supplied image, it is not possible for me to determine whether this is due to non-specific staining or that all thickened septae saturated with neutrophils. Please improve the quality of supplied images.
-We also think that the image was reduced during conversion. We now added the original TIF file and also the original Powerpoint file of the histologies (this is the best resolution we have!) and submitted an additional zip file (Please see our new submission). We hope that these illustrations now meet the requirements.
The presentation of the real-time data in Figure 3 is still confusing for me. To help the reader please expand the material and methods to explain exactly how the “percent of expression 2-ΔΔCT” was calculated. i.e. the following statement is not sufficient “Relative quantification was estimated by the ddCT method [12] with b-actin as a calibrator. The level of target gene expression was calculated using 2-ddCT”. Please state exactly how dCT, ddCT, 2-ΔΔCT and ultimately percent of expression 2-ΔΔCT was calculated. In the analysis of this figure were only -/- and +/+ groups compared? And was “ANOVA with Bonferronni`s Multiple Comparisation Test” used for this comparison as stated in the material and methods section?
-That is absolutely correct. We agree with the reviewer, especially as the description was not fully correct and not complete. The calculation of the efficiency-corrected relative expression was used (therefore: DCT, not DDCT!!). We now added an appropriate and complete description of this calculation method, as well as another reference from the Pfaffl working group dealing with the efficiency-corrected CT model.
-Revised Text:
-Standard curves were prepared for the amplification specific efficiency correction and the efficiencies (E) were calculated according to the equation E = (10−1/m−1) × 100, where m is the slope of the linear regression model fitted over log-transformed data of the input cDNA concentrations versus CT values [12,13]. E for actin-beta was 1,9164 and for luciferase = 1,9399. The relative efficiency-corrected mRNA expression of the target gene was calculated based on efficiencies and the CT (Threshold cycle) deviation of an unknown sample versus a positive control (DCT), and expressed in comparison to a reference gene [12,13]. Data are calculated (rel. expression = (E luc=1,9164)DCT luc / (E actin=1,9399)DCT actin) and expressed as percent (the calculated rel. expression 1,0 refers to 100%, the value 0.02 refers to 2%).
-Nonetheless, if you think the percentage display is confusing, we can change it at any time and change the display of the y-axis to 0 to 1.0, but that doesn't change the results.
-In addition, we have corrected the description of the statistical analysis. Thank you very much for your careful review. Of course evaluations of lung function testing as well as Real-Time PCR were performed using unpaired t-test. We corrected this in the methods section.
Thank you for clarifying that the day 50 timepoint (Figure 4) is from only one mouse.
-We thank the reviewer. Of course, we know that a data point with n=1 is not very meaningful, but nevertheless we wanted to have this data point mentioned in the publication as a preliminary result.
Reviewer 3 Report
The Authors responses satisfy and clarify the reviewer critical points.
This manuscript has been now improved with the addition of new informations that better clarify state of art and Results sustaining the Conclusions.
Author Response
Thank you for your kind and very useful comments.
This manuscript is a resubmission of an earlier submission. The following is a list of the peer review reports and author responses from that submission.
Round 1
Reviewer 1 Report
In the present study, Patrick and colleagues, using a transgenic mouse model, BLI, and q-PCR provide evidence that the in vivo imaging and qPCR results demonstrated migration accompanied by a significant longer retention time of transplanted mASCs in the lung parenchyma of Atm-deficient mice compared to wild type mice. Your study further demonstrated that a combination of luciferase-based PCR, together with BLI, is a pivotal tool for tracking mASCs after transplantation in models of inflammatory lung diseases such as A-T. The manuscript is well written, but there are some unconvincing points.
- This study’s result, Luc+ mASCs stayed longer in lung tissue of Atm-deficient mice compared to wildtype mice is very interesting. As a one researcher, I felt the possibility that this Atm-deficient mice have the niche for MSCs of engraftment and function
- In figure 1, how do you think why were injected mASCs retention longer in Atm-deficient mice compare with wild type mice? Authors discussed that the Atm-deficient mouse has at least some kind of ongoing pulmonary damage. I think the way to resolve this question is not difficult. You have to investigate the histology of this mouse before and after transplantation.
- If you conducted the histological analysis before and after this transplantation model, you’ll get the potential of this therapy for A-T. If this model has the functional analysis, please check the recovery of lung damage.